# CATEGORICAL ENTITY FEATURES IN RECOMMENDATION SYSTEMS USING GRAPH NEURAL NETWORKS

## ABSTRACT

Graph neural networks are widely used in recommender engines and are commonly applied to user-item graphs augmented by various side information, including categorical entity features. It is established that a user selection process involves a complex framework of preferences and the importance of presented alternatives. For example, user's preferences might change depending on product category and/or brand. Thus, comprehending and modeling them effectively is essential in the recommender engines' context. Despite the significant influence of such categorical features on the user decision-making process, these have been incorporated in graph models in various ways without giving a clear indication of which method is most suitable. We investigate the capabilities of graph neural networks to extract and model categorical attribute-specific preferences effectively by systematically comparing existing techniques and graph models. These include one-hot encoding-based node features, category-value nodes, and categories as hyperedges. In addition, we introduce a novel hyperedge-based method designed to leverage categorical features more effectively compared to current approaches. The proposed model, which has a simple architecture and combines neighborhood aggregation with hyperedge aggregations, outperforms many complex and sophisticated methods. In extensive experiments using three real-world datasets, we compare existing methods and demonstrate the advantage of our approach in terms of commonly used quality metrics for recommender engines.

## 1 INTRODUCTION

E-commerce website users encounter the daunting challenge of sifting through an overwhelming number of products to find the right item. To address this issue, recommender system (RS) algorithms have been designed to understand user intentions and predict which items to shortlist. The central objective of these RS algorithms is to learn and extract user preferences effectively, enabling them to anticipate the next likely item of interest. This task poses significant difficulties as users' decision-making process involves quantifying preferences and the importance of presented alternatives (Dyer & Sarin, 1979). For instance, user price preferences are highly influenced by the brand or product category. The process of clicking on the next item is driven by a complex interplay of product attributes and user preferences. Thus, the importance of categorical features of entities is pivotal in effectively learning and modeling user preferences.

Given that user-item interactions can be naturally represented as graph data, where nodes represent users/items and edges correspond to interactions like clicks or purchases, many authors have successfully used graph neural networks (GNN) for recommender engines (He et al., 2020; van den Berg et al., 2017; Li et al., 2023; Sun et al., 2020; Guo et al., 2021; Zheng et al., 2023; Liu et al., 2022; Hu et al., 2020; Li et al., 2021). It is claimed that the advantage of GNN-based user-item recommender systems lies in their ability to incorporate information beyond user-item relations, including edges among users/items and diverse user and item features.

Although GNNs have been adopted for RS, it is noteworthy that there is limited research dedicated to understanding how to incorporate categorical features best and its capability to extract user preferences from such characteristics effectively (e.g., price preferences, brand preference, or interaction of those two).

In this paper, we investigate the role of categorical features in user-item recommender engines based on graph neural networks. We explore various techniques that are used to integrate categorical features of entities. Many papers include such information as binary encoded node features (Sun et al., 2020; Guo et al., 2021) or adding category value-nodes on graphs (Zheng et al., 2023; Liu et al., 2022; Hu et al., 2020; Li et al., 2021). However, authors usually do not explore or clarify why they selected a specific method. There are no definitive guidelines/studies on which approach is most suitable for integration with a particular GNN architecture and whether or not there are other ways to consider. Therefore, we examine existing practices from the literature and propose a new method - category values as hyperedges that demonstrate effective utilization of categorical features compared to current methods. Using hyperedges in recommender engines is not novel and has already been studied (Zhang et al., 2022; Wang et al., 2020; Xia et al., 2021). However, most of the research is focused on session-based recommender engines, where hyperedges are created by combining different attributes together (for example, all prices within sessions build a hyperedge).

It is to be noted that our examination focuses on user-item recommender systems and does not extend to session-based recommender systems. In addition, we concentrate on how entities' categorical features, e.g., users and/or items categorical features, can be effectively utilized and do not study context features, e.g., interactions categorical features.

The main contributions of this paper are as follows

- Examination of categorical feature integration: We review the literature and examine how categorical features are integrated into the models. Furthermore, we extensively compare different techniques to find out how different methodologies impact model performance.

- New architecture: We introduce a new approach where categorical features of entities are used directly as hyperedges in GNN-based user-item recommender engines. We demonstrate that even though our approach has a simple architecture, it surpasses the performance of more sophisticated methodologies.

- Empirical comparison and validation: We conduct extensive experiments on three real-world datasets and show that the hyperedge approach outperforms other methodologies (e.g., category-value nodes and binary-encoded features). In addition, we benchmark our approach against state-of-art models. The findings suggest that hyperedges can effectively be used to extract user preferences that improve model accuracy.

## 2 RELATED WORK

We discuss the related work on categorical features in recommender engines in general and specifically for GNN-based methods.

### 2.1 RECOMMENDER ENGINES USING CATEGORICAL FEATURES

Early recommender systems used only user-item interaction data to generate new recommendations. In this context, categorical features were often considered in the pre and post-processing stages of recommendation generation (Mei et al., 2018; Sun et al., 2019). Several studies implemented item/user categories as pre and post-filters (Hwang et al., 2012; Panniello et al., 2009; Davidson et al., 2010; Baltrunas & Ricci, 2009; Wadhwa et al., 2020). For instance, Davidson et al. (2010), used categories as a post-processing step to further narrow down a subset of items for presentation to the users. Baltrunas & Ricci (2009) utilized contextual item information as a pre-processing step. Pre and post-filters were the first attempts to include additional information in recommender systems.

Advancements in modeling recommendation engines have enabled the integration of categorical features in the learning process. In the context of user-item recommender engines, categorical features are either entity (user/item) specific or user-item interaction specific (Chen et al., 2019). User/item-specific attributes are called side information, for example, user age/gender, item category/brand. On the other hand, user-item interaction-specific features are called context (Meng et al., 2023; Adomavicius & Tuzhilin, 2015). Early studies have explored both context-aware and side information-aware recommender engines and suggested different methods to employ categorical features in the

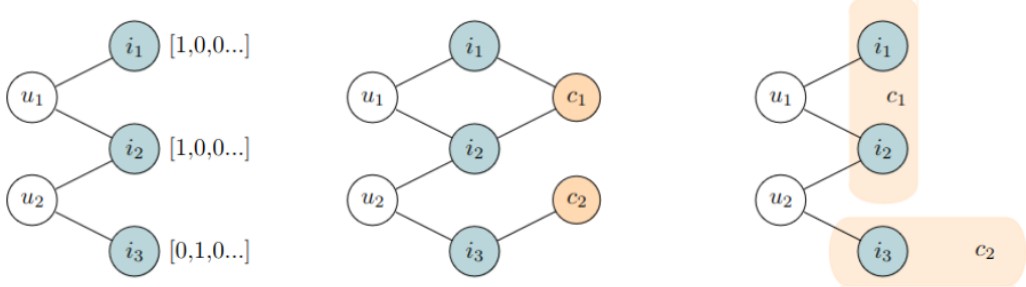

Figure 1: Illustration of three graph models incorporating categorical entity features. In the first graph, categories are considered as features of items by creating binary vectors encoding the categorical value. The second graph represents each categorical value as extra nodes. The graph on the right shows categories as hyperedges.

learning process. In the context of entity categorical features, early latent factor models have utilized them as auxiliary information, serving as sparse features to create a user/item side information matrix (Singh & Gordon, 2008; Veloso et al., 2019; Pasricha & McAuley, 2018).

Representation learning models also leverage user and item features to predict user-item connection (Maeng et al., 2022; Cheng et al., 2016; Covington et al., 2016). These methodologies construct an input feature matrix using dense and sparse user/item features. For example, Dong et el. Dong et al. (2017) constructed user and item feature matrix for the movie lens datasets where item features contain 18 movie genre categories encoded as binary vectors. Similarly, it utilizes the user's age, gender, and occupation.

## 2.2 CATEGORICAL FEATURES IN GRAPH NEURAL NETWORKS

In the absence of rich, distinctive input features for items and users, it is well-established to use the identity matrix of a node as an input features matrix, e.g., each node is described as one hot encoding vector and is unique for every other nodes (He et al., 2020; van den Berg et al., 2017; Li et al., 2023). However, when relevant entity features exist, authors rely primarily on two methods.

The first commonly used technique is constructing binary-encoded vectors to represent categorical values. These binary vectors then are used directly as input features, or they are concatenated with the identity matrix (Sun et al., 2020; Guo et al., 2021). The latter is usually used when entities have insufficient unique features to differentiate users/items.

The second method used is category values as nodes. Several studies have adopted this technique (Zheng et al., 2023; Liu et al., 2022; Hu et al., 2020; Li et al., 2021). For example, Liu et al. (2022) created a use-item-attribute graph. Items were connected to attribute nodes, and user-attribute interest was extracted by an attribute-aware attention mechanism. Similarly, Zheng et al. (2023) included item categorical features (price and categories) as extra nodes on the graph. They designed a two-branch factorization machine to extract price preferences (Sun et al., 2019). Li et al. (2021) utilized item attributes such as categories and location as nodes.

The effectiveness of those methods is not very obvious. For example, some authors have pointed out the limitations of the binary-encoded category method (Zhang et al., 2022; Liu et al., 2022). When included as one-hot encoded features, it becomes very sparse where only a few entries are non-zero, which can lead to learning unreliable parameters (Liu et al., 2022). Similarly, creating category-value nodes and connecting them with item nodes might not directly extract user category preferences and dependences (Zhang et al., 2022).

Furthermore, there is a complex interdependence between the graph model used and the GNN architecture realizing the recommender engines. Various aggregation mechanisms for graphs and hypergraphs have been proposed. Moreover, approaches not only differ in their graph model for categorical features but also use various techniques, such as attention mechanisms, making it difficult

Table 1: Summary of different methods: $|V|$ is number of nodes, $|E|$ is number of edges, M is initial feature vector size, K is number of all categorical values, $C_u$ is number of user category features, $C_i$ number of item category features. $|V_u|$ number of user nodes, $|V_i|$ number of item nodes.

| Method | Order of Graph | Size of Graph | Features |
|---|---|---|---|
| Without categorical features | $|V|$ | $|E|$ | $|V| \times M$ |
| Categories as binary features | $|V|$ | $|E|$ | $|V| \times (M + K)$ |
| Category value nodes | $|V| + K$ | $|E| + (|V_u| \times C_u + |V_i| \times C_i)$ | $(|V| + K) \times M$ |
| Categories as Hyperedges | $|V|$ | $|E| + (|V_u| \times C_u + |V_i| \times C_i)$ | $|V| \times M$ |

to assess the impact of the representation of categorical features, although this is a crucial design decision.

We briefly mention that some authors used categorical features as edge features, mostly in context-aware recommender engines (Wu et al., 2022). Other research papers (Guo et al., 2021) built dual graphs to incorporate attribute information, one for user-item interactions and one for the attributes.

Another way, we suggest, categorical features can be utilized on user-item graphs is to use them directly as hyperedges. In graph theory, hyperedges are edges that connect any number of nodes simultaneously (Yadati et al., 2019; Huang & Yang, 2021). For example, two items can be linked via a hyperedge because they share the same brand and price level.

The concept of hyperedges is not new, and many studies have used hypergraphs and hyperedges to model recommender engines (Zhang et al., 2022; Wang et al., 2020; Xia et al., 2021). However, most studies are limited to session-based recommendation engines, and most importantly, those studies create hyperedges based on combinations of itemID and/or attributes, e.g. they introduce category value nodes into the graph. For example, Zhang et al. (2022) proposed session-based recommender engines, where nodes are price, category, and items. Hyperedges then connect some combination of those nodes, e.g., all price nodes within the session.

The main advantage of hyperedges is that it can naturally model high-order interactions, which is common in real-world scenarios and thus can be utilized to overcome the above-mentioned limitation.

## 3 PRELIMINARIES

### 3.1 GRAPH AND INPUT FORMULATION FOR DIFFERENT TECHNIQUES

Figure 1 depicts three discussed approaches for incorporating categorical features into GNN-based user-item recommender engines, e.g., binary encoding of categorical features, category value nodes, and categories as hyperedges.

Below, we describe how a graph changes when adopting different methods. We define an undirected bipartite graph $G = (V, E)$ with $V$ consisting of user and item nodes $V = V_u \cup V_i$. Edge set $E$ contains interaction edges between user-item nodes $(u, i)$. Users and items have non-categorical feature vectors of size M associated. Let us assume that users and items have $C_u$ and $C_i$ categorical features, respectively. Finally, K is a number of all category values for both user and item. Table 1 summarizes how the order of the graph, size of the graph, and feature matrix transform with different methods. The order is defined as the number of nodes and size as the number of edges (Harris et al., 2008)

We can observe that in the hyperedge method, the size of the graph increases by the number of nodes times category features without increasing the number of nodes or feature matrix. In general, the size of the graph increases by the number of hyperedges. In the case of binary-encoded categorical values, input features grow by the number of all category values. While for category values as nodes, both the graph's size and the graph's order increase, as does the feature matrix.

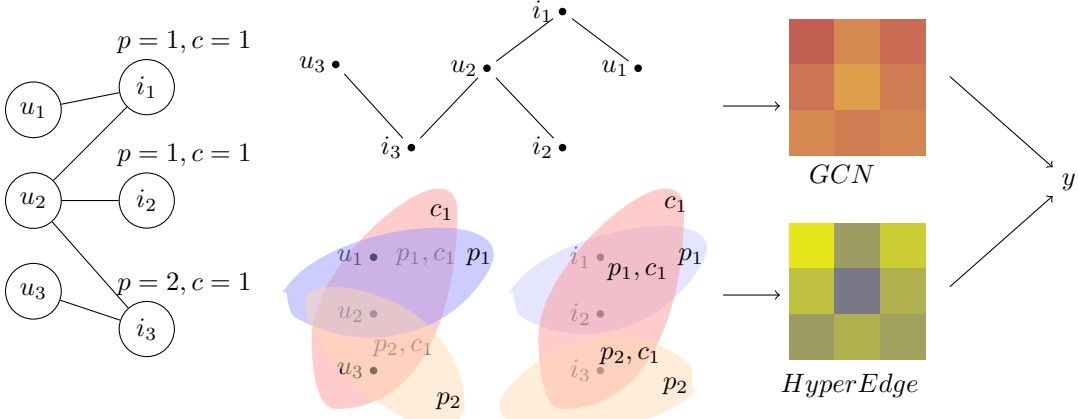

Figure 2: An example of incorporating price level and product category features as hyperedges. As an input, we have a bipartite graph with two types of nodes (users, items), and items have two categorical attributes. For example, $i_3$ has price level 1 and category value 1. On the bipartite graph, we have two types of aggregation. A simple GCN layer that aggregates neighboring information. The second is hyperedge aggregations. Finally, they are combined to make a final prediction.

## 4 METHODOLOGY

In previous sections, we discussed existing methods and motivated our new approach and the concept underlying it. Here, we discuss its concrete realization and present a unified framework to compare the different methodologies. We adopted categories as hyperedge concept for studying price and product category dependency for ecommerece recommender engine.

Figure 2 illustrates the proposed model architecture. Here, we have a standard undirected bipartite graph $G = (V, E)$ with $V$ consists of user and item nodes, $u \in U, i \in I$. Items have two categorical features: $p \in P$ and $c \in C$ (p stands for the price level and c for the product category). Edge set $E$ contains interaction edges $(u, i)$ and hyperedges for each category value $h_c, h_p, h_{cp}$. Hyperedge construction is as follows: For every category value, one hyperedge is created. Then, all items that share the same category value are connected. Similarly, we create hyperedge for all users who interacted with items of the same category value. In addition, interactions hyperedges are constructed $h_{cp}$ (e.g., price level=1 and product category='tablets' is one hyperedge).

During the learning process, we have two types of aggregation on graphs. One is a standard graph convolutional layer (Kipf & Welling, 2017) to capture neighborhoods, and the second is a hyperedge aggregation. Finally, we combine these two aggregations and use them for the prediction. The Pseudo algorithm algorithm is shown in Algorithm 1.

### 4.1 ENCODER

Below is the exact formulation of the encoding part of the model. As mentioned above, for neighborhood aggregation, we use the GCN layer. For the hyperedge aggregation, we adapt the UniSAGE aggregation (Huang & Yang, 2021) extending GraphSAGE (Hamilton et al., 2017) to hypergraphs. The exact node-level formulation for a node $v$ is:

$$h_v^{l+1} = \sigma\left(\left(W_n^l \sum_{u \in N(v) \cup \{v\}} \frac{1}{\sqrt{\tilde{d}_j \tilde{d}_i}} h_u^l\right) \,\Big\|\, \left(W_h^l \left(h_v^l + \sum_{e \in E_v} h_e^l\right)\right)\right), \qquad (1)$$

where the left term corresponds to the node-level formulation of GCN (Kipf & Welling, 2017), $E_v$ is the set of hyperedges containing $v$, $h_e^l$ is the embedding of the hyperedge $e$ obtained as

Table 2: Statistics of the datasets

| Datasets | #users | #items | #interaction | #price level | #category |
|---|---|---|---|---|---|
| Amazon Grocery | 8535 | 12906 | 145755 | 21 | 24 |
| Amazon Tools | 17642 | 26087 | 291361 | 21 | 13 |
| Yelp | 19301 | 17587 | 452931 | 4 | 83 |

$h_e^l = \frac{1}{|e|} \sum_{u \in e} h_u^l$, $W_n^l, W_h^l$ are learnable parameters for neighborhood and hyperedge aggregation, respectively, $\sigma$ is a nonlinear activation function and $\|$ denotes concatenation.

## 4.2 DECODER AND LOSS FUNCTION

To predict user preferences, we use the inner product of the final user and item representations. We combine this with the Bayesian Personalized Ranking (BPR) loss function (Rendle et al., 2009) to train the model. Combination of inner product and BPR loss is a well-established framework for training recommender engines (Yue et al., 2023; He et al., 2020; Liu et al., 2022; Wang et al., 2019; Li et al., 2021; Lin et al., 2022). The exact formulation of the decoder is as follows:

$$y_{ui} = z_u^T z_i$$

where $z_u, z_i$ are the final user item representation. This approach implies that the similarity of a user to an item is proportional to the dot product of their representation (Hamilton, 2020).

BPR loss is a widely used method since it considers positive and negative user-items pairs. BPR encourages models to rank positive user-item interactions higher than negative user-item interactions. The precise formulation of the loss function is as follows:

$$L = \sum_{(u,i,j) \in O} -\ln(\sigma(s(u,i)) - \sigma(s(u,j))) + \lambda \|\Theta\|^2$$

Where $O$ denotes the set of positive-negative sample pairs, representing user $u$ with a positive item $i$ and a negative item $j$, $\sigma$ denotes the sigmoid function, which maps the predicted scores to probabilities between 0 and 1, $s(u,i), s(u,j)$ are predicted scores for positive and negative items, respectively, and $\Theta$ represents the model parameters, where $\lambda$ controls L2 regularization.

## 5 EXPERIMENTATION

### 5.1 EXPERIMENTAL SETTINGS

**Research Questions:** In our study, we performed extensive experimentation to evaluate various approaches and answer the following research questions:

- **RQ1** Do existing GNN-based user-item recommendation systems benefit from categorical entity (user/item) features?
- **RQ2** What is the best way to incorporate categorical features in a graph model?
- **RQ3** Can we develop GNN-based user-item recommender engines effectively using categorical features to improve their prediction accuracy?

**Datasets:** To examine model performances, we use three real-world data sets: Yelp2018, Amazon Tools 5core, and Amazon Grocery 5core datasets. Table 2 depicts a summary of datasets.

- Yelp2018[1] dataset is widely used for recommender engines. Here, restaurants are considered as items for which users have reviews. Price categories, e.g., how expensive the restaurant is, and restaurant subcategories are extracted. We follow the same approach as the PUP paper and use a 10-core setting, only keeping users and items with at least ten interactions.

---

[1] https://www.yelp.com/dataset/

- Amazon Tools 5 core[2] is adapted. Subcategories and prices are used to create categorical features. Price buckets are created by grouping values within an interval of 5. Furthermore, subcategories are used to create category features. The Same as above, we apply 10-core settings.
- Amazon Grocery 5 core[3] similar to Amazon Tools dataset we use subcategories and prices. Price categories are created by grouping prices into 5-euro buckets. The first-level subcategories are used as categories. The same as the above 10-core setting is applied.

For each dataset, we rank the interactions by timestamps. We then split consecutively 60/20/20 as training, validation, and testing datasets. We use 1:1 negative sampling, e.g., for every positive training edge, we create one negative sample. Item is considered negative if a user did not interact with it.

**Evaluation Metrics:** To evaluate the model performances, we adapted two widely used evaluation metrics, Recall at K and Normalized Discounted Cumulative Gain (NDCG) at K position (He et al., 2015). Recall@K measures how many items are in the top-K recommended items, while NDCG@K focuses on the quality of the ranking. NDCG@K takes into account the position in which item was recommended. We used 50 and 100 top-K ranks. The reported results are average values over the number of users. Furthermore, we run each method 10 times and mean values are reported in the tables.

**Baselines:** To answer RQ1, we construct different variations of the same model where only the input is different, e.g., The categorical features are added either as binary encoded input features or we create category-value nodes on the graph or using them as hyperedges. In addition, one extra model is constructed as a complete baseline where no categorical features are included, only relying upon the user-item identity feature matrix as input features. We describe the exact model formulations.

In the methodology chapter, we described in detail how $GCN_h$ is aggregated. For all other methods in RQ1, we use a simple GCN layer for the model encoding part, e.g., we use the first part of the equation 1, followed by the activation function. The model prediction and training process is identical to $GCN_h$, which is described in the decoder and loss function section.

- $GCN_w$ $\mathbf{F} \in \mathbb{R}^{n \times n}$ input is the user-item identity feature matrix, where $n$ is the number of nodes.
- $GCN_n$ $\mathbf{F} \in \mathbb{R}^{n+c \times n+c}$ considers categorical values as extra nodes on the graph. e.g., the size of the input matrix is increased by a number of categorical values.
- $GCN_f$ $\mathbf{F} \in \mathbb{R}^{n \times n+c}$ adds categorical values in the feature matrix.
- $GCN_h$ $\mathbf{F} \in \mathbb{R}^{n \times n}$ does not increase the size of the input features matrix but uses categorical features for hyperedge construction.

In each dataset, there are two category features: price level and product category. For RQ1, we test three scenarios per dataset, e.g., only price level, only product category, and both together price level and category. Hence, we have nine different frameworks to compare in total.

To test RQ3, we compare our hyperedge approach with the state-of-the-art models. The competitive models we picked are BPR-MF, A2-GCN, PUP, and CatGCN. All except BPR-MF are incorporating categorical features into the model learning process.

- **BPR-MF** Koren et al. (2009) is a classical matrix factorization method combined with Bayesian personalized ranking loss for optimization. It is only based on user-item interactions and ignores side information.
- **A2 GCN** Liu et al. (2022) is an attribute-aware recommender engine that incorporates categorical attributes as extra nodes in the graph. It uses an attention mechanism to model user preferences.
- **PUP** Zheng et al. (2023) is price aware recommender engine. This method considers categories as nodes and deploys a custom decoder to capture the global and local influence of prices and categories.

---

[2]https://cseweb.ucsd.edu/~jmcauley/datasets/amazon_v2/
[3]https://cseweb.ucsd.edu/~jmcauley/datasets/amazon_v2/

Table 3: Performance comparison with different approaches to include categorical features at K=50

| Dataset | Model | Price | | Category | | Price And Category | |
|---|---|---|---|---|---|---|---|
| | | Recall@50 | nDCG@50 | Recall@50 | nDCG@50 | Recall@50 | nDCG@50 |
| Amazon Grocery | $GCN_w$ | 0.0745 | 0.0342 | 0.0745 | 0.0342 | 0.0745 | 0.0342 |
| | $GCN_n$ | 0.0769 | 0.0352 | 0.0751 | 0.0342 | 0.0782 | 0.0357 |
| | $GCN_f$ | 0.0728 | 0.0328 | 0.0720 | 0.0328 | 0.0700 | 0.0317 |
| | $GCN_h$ | **0.0802** | **0.0370** | **0.0813** | **0.0377** | **0.0822** | **0.0377** |
| Amazon Tools | $GCN_w$ | 0.0321 | 0.0139 | 0.0321 | 0.0139 | 0.0321 | 0.0139 |
| | $GCN_n$ | 0.0346 | 0.0150 | 0.0320 | 0.0139 | 0.0342 | 0.0149 |
| | $GCN_f$ | 0.0307 | 0.0132 | 0.0306 | 0.0131 | 0.0289 | 0.0124 |
| | $GCN_h$ | **0.0383** | **0.0164** | **0.0379** | **0.0165** | **0.0383** | **0.0166** |
| Yelp | $GCN_w$ | 0.2137 | 0.0984 | 0.2137 | 0.0984 | 0.2137 | 0.0984 |
| | $GCN_n$ | 0.2157 | **0.1001** | 0.2138 | 0.0983 | 0.2158 | 0.1003 |
| | $GCN_f$ | 0.2137 | 0.0983 | 0.2133 | 0.0979 | 0.2136 | 0.0980 |
| | $GCN_h$ | **0.2150** | **0.1001** | **0.2172** | **0.1011** | **0.2204** | **0.1024** |

- **CatGCN** Chen et al. (2023) approach uses item categorical side information to enrich initial user feature representation. CatGCN is implemented for user node classification tasks. We adopt this approach for link prediction tasks. To adapt this approach for the link prediction, we do as follows: we use items categorical features to enrich users' initial representation. In the case of item features, we adopt the identity matrix. We then combine user and item features and pass them into GCN layers. The training process for the link prediction is identical to our hyperedge approach, e.g., we use the same decoder mechanism.

**Implementation Details:** For all baselines, we used the publicly available original implementations with their default parameters. We set the maximum epoch for training to 200. For our hyperedge model, we did hyperparameter search for the learning rate in (0.1, 0.01, 0.001, 0.0001) and L2 normalization in (1e-10, 1e-8, 1e-5, 1e-4) using the BPR loss function. The embedding size is fixed for 64. Adam optimizer is used for the optimization. The training happens in full batch mode. We use a one-layer model and report average values over ten runs.

## 5.2 Performance Comparison RQ1 and RQ2

Table 3 shows model performances at top-K=50 position. In the table, we highlight in bold the best performances. There are several interesting observations. First, we see that adding categorical features to the model is not always beneficial. In all datasets $GCN_w$ is better than $GCN_f$. This is contrary to the expectation that more features in the model the better. This does not necessarily mean that features are meaningless. Rather, it could be that the model cannot learn reliable parameters for sparse input features.

Including categorical values as extra nodes is usually better than not including them at all. In 7 out of 9 scenarios, $GCN_n$ is better than $GCN_w$. Furthermore, the results show that for almost all cases, including categorical features as nodes is superior to the binary-encoded method.

The second research question focuses on identifying the best way to include categorical features. Our results suggest that including category features as hyperedges is always better than not including them at all, and by large, the hyperedge method outperforms other methods in almost all scenarios. Only in one case, $GCN_n$ has better results than $GCN_h$.

Furthermore, performance varies across different datasets, indicating that the efficacy of model selection is influenced by dataset structure.

The results for top-K=100 can be found in the Appendix. We make similar observations as in top-K=50. This finding suggests that models generally do not necessarily and automatically benefit from categorical features. And it should be part of the model selection to decide how to integrate categorical features.

Table 4: Performance comparision with competitive baselines

| Datasets | Model | Recall@50 | Recall@100 | nDCG@50 | nDCG@100 |
|---|---|---|---|---|---|
| Amazon Grocery | BPR-MF | 0.0569 | 0.0834 | 0.0276 | 0.0337 |
| | CatGCN | 0.0349 | 0.0607 | 0.0139 | 0.0199 |
| | A2-GCN | 0.0510 | 0.0853 | 0.0212 | 0.0291 |
| | PUP | 0.0745 | 0.1106 | 0.0340 | 0.0424 |
| | $GCN_h$ | **0.0822** | **0.1209** | **0.0377** | **0.0467** |
| Amazon Tools | BPR-MF | 0.0282 | 0.0443 | 0.0123 | 0.0160 |
| | CatGCN | 0.0123 | 0.0232 | 0.0047 | 0.0073 |
| | A2-GCN | 0.0236 | 0.0404 | 0.0097 | 0.0135 |
| | PUP | 0.0321 | 0.0511 | 0.0140 | 0.0184 |
| | $GCN_h$ | **0.0383** | **0.060**9 | **0.0166** | **0.0218** |
| Yelp | BPR-MF | 0.2123 | 0.3280 | 0.0999 | 0.1291 |
| | CatGCN | 0.1054 | 0.1831 | 0.0462 | 0.0663 |
| | A2-GCN | 0.1883 | 0.2979 | 0.0889 | 0.1167 |
| | PUP | **0.2221** | **0.3417** | **0.1024** | **0.1326** |
| | $GCN_h$ | 0.2204 | 0.3384 | **0.1024** | **0.1322** |

## 5.3 PERFORMANCE COMPARISON RQ3

In RQ1 and RQ2, we were solely interested in understanding if there is any difference in how categorical features are included in GNNs. Hence, we used standard GCN approaches to compare various techniques.

To answer RQ3, we further compare the hyperedge model with current state-of-the-art models. Table 4 summarizes the experiment results and shows that our approach is, by large, the most effective way to model categorical features with PUP having competitive results. The model performance of $GCN_h$ is particularly strong in the Amazon Grocery and Amazon Tools dataset. The Amazon Grocery dataset $GCN_h$ outperforms second-best results by 10 percent. In Amazon Tools, improvement is almost 18 percent compared to second-best results. In the Yelp dataset, our model has competitive performance. It is notable that in some cases even simple BPR-MF outperforms competitive baselines such as A2-GCN and CatGCN.

The hyperedge model has the simplest architecture compared to A2-GCC, PUP, and CatGCN, which rely on attention mechanisms, customized decoder, or local and global embedding learnings. Still, our approach outperforms those methods and, in some cases, has a significant margin.

## 6 CONCLUSIONS AND FUTURE WORK

This research paper examined different methods to incorporate categorical features of entities into GNN-based user-item recommender engines. Extensive experimentation was conducted to compare traditional approaches, such as category-value nodes and binary-encoded category features, to category-value hyperedges, as well as using no categorical features at all. We tested in three datasets with three different scenarios (e.g., including only product category, price level, or both of them together). Our findings suggest that the hyperedge approach outperforms other techniques in all cases. Another interesting observation is that including categorical binary-encoded features makes the model almost always worse than not including them at all. Furthermore, we compared the hyperedge approach to competitive baselines such as PUP, A2-GCN, and CatGCN, which studied categorical features in GNN-based user-item recommender engines. By large, the findings demonstrate the superiority of the hyperedge approach.

For future work, further investigation is needed into how model architecture influences the most effective method for incorporating categorical features. Moreover, we hope that our study will motivate other researchers to dive deep into GNNs' ability to extract complex user preferences as well as category dependencies.

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

# A   APPENDIX

Table 5: Performance comparison with different approaches to include categorical features at K=100

| Dataset | Model | Price | | Category | | Price And Category | |
|---|---|---|---|---|---|---|---|
| | | Recall@100 | nDCG@100 | Recall@100 | nDCG@100 | Recall@100 | nDCG@100 |
| Amazon Grocery | $GCN_w$ | 0.0745 | 0.0342 | 0.0745 | 0.0342 | 0.0745 | 0.0342 |
| | $GCN_n$ | 0.1144 | 0.0439 | 0.1122 | 0.0428 | 0.1153 | 0.0443 |
| | $GCN_f$ | 0.1098 | 0.0414 | 0.1074 | 0.0410 | 0.1058 | 0.0400 |
| | $GCN_h$ | 0.1191 | 0.0461 | 0.1204 | 0.0468 | 0.1209 | 0.0467 |
| Amazon Tools | $GCN_w$ | 0.0321 | 0.0139 | 0.0321 | 0.0139 | 0.0321 | 0.0139 |
| | $GCN_n$ | 0.0552 | 0.0197 | 0.0515 | 0.0184 | 0.0547 | 0.0196 |
| | $GCN_f$ | 0.0493 | 0.0175 | 0.0496 | 0.0175 | 0.0470 | 0.0165 |
| | $GCN_h$ | 0.0608 | 0.0216 | 0.0604 | 0.0216 | 0.0609 | 0.0218 |
| Yelp | $GCN_w$ | 0.2137 | 0.0984 | 0.2137 | 0.0984 | 0.2137 | 0.0984 |
| | $GCN_n$ | 0.3325 | 0.1296 | 0.3301 | 0.1277 | 0.3327 | 0.1298 |
| | $GCN_f$ | 0.3306 | 0.1279 | 0.3299 | 0.1274 | 0.3297 | 0.1274 |
| | $GCN_h$ | 0.3316 | 0.1296 | 0.3347 | 0.1308 | 0.3384 | 0.1322 |

---

**Algorithm 1** An algorithm

    **Input:** $G = (V, E)$, $L = 200$
    **Output: Z**
Initialize model parameters
Construct **A** adjacency matrix for neighbourhood
Construct hyperedges $h_c^u, h_p^u, h_{cp}^u, h_c^i, h_p^i, h_{cp}^i$ for users and items respectevly
    **for** $i = 1, \ldots, L$ **do**
        Obtain $X^n$ using $GCN$ layer for neighbourhood aggregation
        Obtain all $X_k^j$ where $j \in \{u, i\}, k \in \{c, p, cp\}$ by hyperedge convolution
        Obtain $X^{hyper}$ by summing all hyperedge convolutions
        Obtain final node embeddings **Z** by concatinating $X^n$ and $X^{hyper}$
        Update model parameters by loss function
    **end for**
Return final node Embeddings **Z**

