# OpenReview forum: "Categorical Features of entities in Recommendation Systems Using Graph Neural Networks"
_ICLR.cc/2024/Conference — Submitted to ICLR 2024_

### Official Review · Reviewer_FdC6 · 2023-10-31

**Soundness:** 2 fair
**Presentation:** 3 good
**Contribution:** 1 poor
**Rating:** 3
**Confidence:** 4

**Summary:**

This paper focuses on how to efficiently model categorical attributes within user-item graph networks for recommender systems. The authors first compare existing methodologies based on 1) one-hot-encoding of binary features for the entities, 2) creation of nodes representing an attribute linked to entities possessing the attribute and 3) creation of hyperedge between any entities sharing the same attribute. Then, they proposed a new model where categorical attributes were handled as hyperedges.

**Strengths:**

Different ways of handling the categorical features in GNN are well described.
Experiments are made on 3 real-world datasets.
The hyperedge trick to handle categorical features seem to provide the best results according to Table 1 for the 3 tested datasets: Amazon Grocery, Amazon Tools and Yelp.

**Weaknesses:**

This is clearly stated by the authors, the proposed approach is only for user-item recommender systems and not session-based ones and does account for item or user features independently only, not interaction categorical features.

By the way, I think the paper would benefit from a quick explanation on why extension of the concept of hyperedges for session-based recommender systems to user-item recommender systems is not straightforward, to justify the novelty of the approach here (which is unclear to me).

“it is noteworthy that there is limited research dedicated to understanding how to incorporate categorical features best“. However, reading the related work, it seems rather clear the limitations of one-hot-encoding and attribute nodes: “some authors have pointed
out the limitations of the binary-encoded category method“.

“For all baselines, we used the publicly available original implementations with their default parameters.” I don’t think this is the correct way to proceed. Each baseline needs to be optimized for the use case for fair comparison.

Some comments:

-In Equation (1), \tilde{d}_j, \tilde{d}_i  are not defined and I would also mention that N(v) stands for the neighborhood of node v.

-z_u and z_i are not mathematically defined in p.5.

Minor, typos:

p.3, “use-item-attribute graph”.

p.4. “Another way…” sentence needs to be rewritten.

p.7 “each datasets”

**Questions:**

Can you please explain why the extension of the concept of hyperedges for session-based recommender systems to user-item recommender systems is not straightforward, to justify the novelty of the approach here?

How are the experimental results after tuning the hyperparameters of each competing approach?

Did you study the benefit of the approach with more categorical attributes and not only price level and brand category to describe the items?

What is the effect of the number of different valuations for each categorical feature?

=== AFTER REBUTTAL ===

I thank the authors for taking the time to answer our questions. Unfortunately I don't upgrade my score because I think the novelty is too limited and the proposed setup (with 2 types of categorical features) too restrictive.

---

### Official Review · Reviewer_p2X9 · 2023-11-03

**Soundness:** 2 fair
**Presentation:** 3 good
**Contribution:** 1 poor
**Rating:** 3
**Confidence:** 4

**Summary:**

The authors studied the problem of how to properly incorporate categorical features into graph neural models. The authors' contribution is mainly in two folds:
The authors compared with multiple commonly used baselines to estimate which way to incorporate categorical features worked better
The authors proposed a new model to represent categorical features as hyper edges in the graphs.

**Strengths:**

Strength
- The paper is in general well written and easy to follow

- The experiments are conducted on 3 public dataset which is easy to follow and repeat the experiment

**Weaknesses:**

Concerns
- My major concern is the lack of technical contribution. As pointed out by the authors, using hyperedges in recommender engines is a very straightforward idea and is not novel. The authors' argument of  "It is to be noted that our examination focuses on user-item recommender systems and does not extend to session-based recommender systems." Does not justify well for the novelty or technical contribution of this paper, which leads to my major concern.

- The author only compared 2 commonly used ways of representing categorical features which is far from being comprehensive, which further decrease the contribution of the paper.

**Questions:**

When building the hyper edges for the proposed model, the authors used secondary interaction between categorical features. What's the time and storage complexity for the proposed algorithm? Will it explode the system if there are a lot of categorical features available for the users and items?

---

### Official Review · Reviewer_t3i6 · 2023-11-03

**Soundness:** 3 good
**Presentation:** 2 fair
**Contribution:** 1 poor
**Rating:** 3
**Confidence:** 4

**Summary:**

This study investigates the integration of categorical features of items into collaborative filtering. Its main idea involves connecting nodes with same attributes through hyperedges and leveraging a hypergraph neural network for encoding. The proposed model is tested across three publicly available datasets, demonstrating a notable improvement over existing benchmarks.

**Strengths:**

1. The paper studies one important task, i.e., collaborative filtering with item attributes.
2. Experiments are conducted on three public datasets.
3. Experiments show that the proposed method outperforms several existing baselines.

**Weaknesses:**

1. Limited novelty. The idea of connecting nodes that share the same attributes with hyperedge is not new and has been explored [1]. As a result, the whole paper seems to be a straightforward application of it on collaborative filtering task, limiting the novelty. Although the performance is promising, providing new insights for the community could be more important for an academic paper.
2. The proposed method considers only two specific attributes, i.e., price and category, which makes the model less generalizable. It could be better if the proposed method describes how it will deal with general single attributes and multiple attributes (e.g., will it be better if we model some combinations of attributes?).
3. Code is not available, making it difficult to reproduce the work in the reviewing phase.
4. The paper lacks robustness in its experimental validation, as there is no evidence of repeated experiments or statistical tests such as paired t-tests, which are crucial for ensuring the reliability of the results.
5. Presentation issue. Figure 2 is not a vector graph and is in low resolution.


[1] Wu et al. Dual-view hypergraph neural networks for attributed graph learning. Knowledge-Based Systems 2021.

**Questions:**

For the "Pricae and Category" setting in Table 2, do you use only $h_{cp}$ or use $h_{cp}$, $h_c$, and $h_p$?

---

### Official Review · Reviewer_ziNY · 2023-11-06

**Soundness:** 2 fair
**Presentation:** 3 good
**Contribution:** 3 good
**Rating:** 6
**Confidence:** 4

**Summary:**

The paper proposes to leverage the categorical information of items for graph neural networks-based recommendation. Differently from previous similar approaches in the literature, the authors do not address the task of session recommendation, which is the main scenario where categorical information is usually injected in recommendation. Specifically, the paper introduces three possible variants of graph-based recommender systems exploiting the categorical information, namely: 1) one-hot encoding of the categorical information as items’ node features, 2) tripartite graphs where categories represent another type of nodes besides users and items, and 3) items’ categories regarded as hyperedges. The authors’ proposal involves the latter setting, where a neighborhood and hyperedge aggregation are performed through a GCN layer and a UniSAGE aggregation, respectively. Finally, the loss function is the common Bayesian personalized ranking one (i.e., BPR). The proposed approach is tested against three GCN architectures having: 1) no categorical information, 2) category information as extra nodes in the graph, and 3) category information as extension of the node features. The evaluation is run on three popular recommendation datasets which include two types of category accounting for either the products’ price or category or both. Results on all such settings demonstrate the efficacy of the hyperedge-based solution, whose trends are further confirmed by evaluating the proposed model against similar state-of-the-art recommendation approaches.

**Strengths:**

+ The proposed approach is simple.
+ The adoption of hyperedges in graph-based recommendation is quite recent in the literature.
+ The authors outline the differences with respect to the existing literature in a sufficient manner.
+ A wide range of evaluation settings are proposed.

**Weaknesses:**

- While proposing a simple approach is not generally criticisable, it might need further discussions regarding the actual novelty of the solution.
- No code is released at review time; this might have been helpful to further assess the efficacy and effectively of the proposed approach.
- Some evaluation choices are not common in the literature and require further justifications.

**After the rebuttal.** The answers provided by the authors addressed the outlined weaknesses quite sufficiently.

**Questions:**

* Can the authors further elaborate on why the proposed approach should represent a novelty to the existing similar approaches? Indeed, it seems that the presented solution makes use of other graph neural networks layers without any specific new techniques introduced.
* Is there any specific reason why the recommendation metrics are calculated with high cut-offs (i.e., at 50-100)? Did the authors try to evaluate the recommendation performance at 10-20? And if so, are the observed trends still confirmed?

**After the rebuttal.** The answers provided by the authors answered my questions quite sufficiently.

---

### Author Response · Authors · 2023-11-22

**Regarding the novelty:**

The novelty of the approach is to use categorical features more effectively to capture category interdependence in the decision-making process. Below are some important points.
- Item's feature is used to create a hyperedge for users. This is not done in any other paper.
- We introduce feature interaction hyperedges to capture dependencies. This way, expert knowledge can be included in the model architecture. In experiments, we demonstrated that this simple method outperforms complex architectures used in the state-of-the-art baselines.
- In the recommender engines context, we are not aware of any paper that uses categorical features directly as hyperedges.
- Session-based recommender engines have adopted the concept of hyperedges, but not in the way we are proposing. In SB recommender engines, a hyperedge can be, for example, all price nodes with which the user interacted (e.g category value nodes are added to the graph) or all items in a session.
- One of the main intentions of the study is to review and compare different methods that have not been done before.

**Regarding the top K:**

50- and 100-cutoff points were used in one of the main papers [1] we used as the baseline. We followed their implementation closely, hence the choice of 50-100. The results are also true in the top 10 and 20 cases.

**Regarding Code:**

We will publish our code on GitHub together with a non-anonymised version of the paper at a later stage. We are sorry for not providing a version of our code with this submission.

**Regarding the Referenced paper:**

There are several major differences with regard to the referenced paper [2].
- The provided reference concerns the node classification task, in which attributes can be any attribute without distinction to categorical features.
- The reference paper considers one type of node that connects attributes common in node features. We have different types of nodes and use features of item node type to build user hyperedges. This can go both ways. One can create item hyperedges based on users' categorical features.
- We introduced feature interaction hyperedges that make it possible to include expert knowledge about the dependencies. As we pointed out, there is overwhelming evidence that price and category are intertwined.
- We tried to keep the architecture simple to understand if the model performs better because of the proposed way categorical features are included.

**Regarding Robustness:**

Every test was run ten times, and we reported mean values. This approach is commonly used in the evaluation of recommender engines. We have clarified this in the updated manuscript.

**Regarding Presentation:**

We have updated the figure.

**Regarding SB recommender engines:**

Session-based recommender engines have adopted the concept of hyperedges, but not in the way we propose. In SB recommender engines, a hyperedge can be, for example, all price nodes with which the user interacted (e.g category value nodes are added to the graph) or all items in a session. We are not aware of any SB recommender engines approach where they have directly categories as hyperedges.

**Regarding the number of methods:**

We have done a comprehensive literature review to pick these methods. We would be happy if you could point out what other method we should consider.

**Regarding the Interactions term:**

We did not intend interaction terms to be general, e.g., if we have ten categorical features and create interaction terms for each of them. Rather, we suggest that expert knowledge can be included too. Many studies show that price and category are highly dependent. That's the reason we included interaction terms. In experiments, we showed that explicitly including this dependency in a simple manner outperforms complex architectures used in the state-of-the-art baselines.

[1] Zheng et al. Incorporating price into recommendation with graph convolutional networks. IEEE Trans. Knowl. Data Eng. 2023

[2] Wu et al. Dual-view hypergraph neural networks for attributed graph learning. Knowledge-Based Systems 2021.

---

> ### Comment · Reviewer_ziNY · 2023-11-22
>
> Dear Authors,
>
> thank you for your careful answers to our reviews. I'll comment on your answers regarding my outlined weaknesses and questions.
>
> - **Novelty.** The points you outlined are now sufficient to justify the novelty of the proposed approach. Thank you.
> - **Top-k evaluation.** Despite it would have been useful to actually take a look at the top-10/20 results (maybe you might include them in the appendix) it is reasonable that you used top-50/100 evaluation to be coherent with one of the baselines.
> - **Code sharing.** Thank you for the clarification, but I'm still convinced the code would have been useful at review time to better assess the implementation of your model and check on the reproducibility. In case of acceptance, please make sure to include it in the final publication.
>
> In the light of above, I'll raise the rating for the paper to a weak acceptance.

---

### Meta-Review · Area_Chair_EtsH · 2023-12-06

**Metareview:**

This paper first studies the scenario of using categorical features for GNNs-based recommendations. Complementarily, the authors propose a hyper-edge based approach to leverage such features more effectively within the GNN.

Key strengths:
* Motivation for studying this problem and proposing this approach is convincing
* The method is simple

Key weaknesses:
* Novelty is cited as an issue by 3/4 reviewers
* The paper considers only two types of categorical attributes, limiting the overal contribution and significance
* The study is not as comprehensive as claimed, since only two ways of categorical feature representation is used for comparison

Mixed:
* The reviewers are generally satisfied with experiments, although RFdC6 makes a good point about ensuring fair comparison with other methods, i.e. either not rely blindly on default parameters or comparing with published results.

**Justification For Why Not Higher Score:**

There is one rebuttal text addressing all reviewers. I find the overall rebuttal rather limited and the arguments within it rather unconvincing. Although one reviewer raised their score, the pool of reviewers as a whole generally feels that post-rebuttal their concerns have not been addressed adequately.

**Justification For Why Not Lower Score:**

N/A

---

### Decision · Program_Chairs · 2024-01-16

Reject